# Efficient Training of Retrieval Models Using Negative Cache

**Erik M. Lindgren**
Google Research, New York
erikml@google.com

**Sashank Reddi**
Google Research, New York
sashank@google.com

**Ruiqi Guo**
Google Research, New York
guorq@google.com

**Sanjiv Kumar**
Google Research, New York
sanjivk@google.com

## Abstract

Factorized models, such as two tower neural network models, are widely used for scoring (query, document) pairs in information retrieval tasks. These models are typically trained by optimizing the model parameters to score relevant "positive" pairs higher than the irrelevant "negative" ones. While a large set of negatives typically improves the model performance, limited computation and memory budgets place constraints on the number of negatives used during training. In this paper, we develop a novel negative sampling technique for accelerating training with softmax cross-entropy loss. By using cached (possibly stale) item embeddings, our technique enables training with a large pool of negatives with reduced memory and computation. We also develop a streaming variant of our algorithm geared towards very large datasets. Furthermore, we establish a theoretical basis for our approach by showing that updating a very small fraction of the cache at each iteration can still ensure fast convergence. Finally, we experimentally validate our approach and show that it is efficient and compares favorably with more complex, state-of-the-art approaches.

## 1 Introduction

Learning to represent objects as dense vectors, often called *embeddings*, has proved to be crucial in large scale information retrieval tasks from multiple domains including recommendation systems [39], vision [18] and natural language processing [13, 22]. A popular paradigm for such learning tasks involves training two separate neural networks (often called two-towers or dual-encoders), each representing a query and a document. Given positive and negative (query, document) pairs, the learning task trains the two networks by minimizing a loss function, usually softmax cross-entropy, to encourage positive pairs to have higher similarity scores and negative pairs to have lower scores. While it is easy to sample positive pairs of examples through user feedback such as impressions or clicks, it is more challenging to sample good negative pairs from a pool of potentially millions or even billions of documents. A large number of negative pairs is often required to ensure high quality of the final model, which makes the training process expensive.

A number of strategies have been proposed in the literature to address the problem of sampling good negative pairs from a large corpus. The most common approach is to use *in-batch* negatives, which treats random, non-positive pairs in a minibatch as negatives [15, 22]. This approach is computationally efficient and works in a streaming setting, but the pool of negative examples is limited to the minibatch. Towards the later stages of the training, the *in-batch* negatives become less informative (i.e., have low gradients) since they are sampled randomly without paying attention

35th Conference on Neural Information Processing Systems (NeurIPS 2021).

to which negatives are hard for a given query [22]. Another popular approach is to maintain an asynchronous retrieval index of the *full dataset* for negative sampling [13, 38]. Negatives from the full dataset can be extracted based on approximate retrieval techniques such as ScaNN [11], Faiss [20] or SPTAG [7]. However, it requires coordinating with a separate process for re-indexing and re-building the retrieval index, which is not only computationally expensive and hard to maintain but also suffers from the problem of stale index.

In this work, we theoretically and experimentally analyze training dual encoders using a large cache of negative elements. The cached elements are *stale*, as they may be generated from a prior iteration's parameters. Our contributions are the following.

**Main Contributions:**

- We propose an approach to train retrieval models with cross-entropy loss using a large negative cache. We utilize Gumbel-Max sampling on the cached embeddings to efficiently sample the negatives.

- We analyze the convergence of our algorithm in terms of the *refresh rate* of the cache. We show that even for a small refresh rate we can obtain a first-order convergence rate comparable to that of getting exact gradients using the entire dataset.

- We develop a streaming version of our approach. Our streaming algorithm allows us to scale to very large datasets and avoids needing to maintain an up-to-date index for nearest neighbor search (or maximum inner product search). We analyze the bias induced by our method and how it affects the convergence.

- We experimentally validate our method using the MS MARCO and TREC 2019 passage retrieval tasks. We show that our approach can be efficiently implemented and achieves statistical performance comparable to state-of-the-art benchmarks with a computationally simpler approach that requires only a fraction of the memory.

## 2 Background

### 2.1 Dual Encoders

Given a query $q$, our goal is to match the query with the most relevant documents. The set of documents is represented by $\mathcal{D} = \{z_1, z_2, \ldots, z_m\} \in \mathbb{R}^{d_D}$, where $m$ is the total number of documents. While we refer to them as documents in this paper, they can be any set of items that we wish to retrieve e.g. movies or products.

The training data consists of $n$ positive (query, doc) pairs $\mathcal{T} = \{(q_1, y_1), (q_2, y_2), \ldots (q_n, y_n)\}$, where $q_i$ is a feature vector in $\mathbb{R}^{d_Q}$ and the label $y_i$ is an integer in $[m]$, which indicates a relevant document for query $q_i$. Our goal is to learn a *dual encoder* model [6] for embedding the queries and documents. Formally, a dual encoder consists of two components:

1. **Query encoder** $\phi_Q : \mathbb{R}^{d_Q} \mapsto \mathbb{R}^d$, which maps a query to a $d$-dimensional embedding space.

2. **Document encoder** $\phi_D : \mathbb{R}^{d_D} \mapsto \mathbb{R}^d$, which maps a document to the embedding space.

The score for a particular query, document pair is simply computed as inner product of the corresponding query and document embeddings i.e., $s(q, z) = \phi_Q(q) \cdot \phi_D(z)$. Intuitively, this represents the "similarity" between the query and documents in the embedding space. The model is trained such that the relevant query, document pairs have high scores. Dual encoders are popular in large scale settings since they are highly efficient during training and inference as the high scoring items for a given query can be found using efficient nearest neighbors algorithms, often achieving more than 100x speed up over brute force search [10].

### 2.2 Training Dual Encoders with Cross-Entropy Loss

In this paper, we focus on the popular cross-entropy loss for training dual encoders. Even though these loss functions have been found to be highly effective, the main challenge in large scale settings

is computational. To observe the computational bottleneck, we look at the cross-entropy loss:

$$\mathcal{L}_{\mathrm{CE}_i}(\theta) = -\log\left(\frac{\exp(\beta s_{iy_i})}{\sum_{j=1}^{m}\exp(\beta s_{jy_j})}\right), \tag{1}$$

where $\beta$ is the *inverse temperature* that scales the scores. Here, $s_{ij} = s_{ij}(\theta) = \phi_Q(q_i;\theta)\cdot\phi_D(z_{y_i};\theta)$ and $\theta \in \mathbb{R}^p$. This computation involves computing the scores for all documents, which can be prohibitively expensive when the number of documents $m$ is large. For simplicity, throughout this paper, we will assume that the embeddings have unit bounded norms, i.e., $\|\phi_Q(\cdot;\theta)\|_2 \leq 1$ and $\|\phi_D(\cdot;\theta)\|_2 \leq 1$ for all $\theta \in \mathbb{R}^p$. Note that this can be assumed without a loss in generality since one can adjust the inverse temperature $\beta$ accordingly to compensate for the scale.

On large-scale datasets, it may not be possible to have the entire dataset in memory, especially with large embedding models where we need the activations of the intermediate layers to persist for back-propagation. An alternative approach is to first select a sample of documents $S$ and use them to approximate the partition function. We thus have the following loss function:

$$\mathcal{L}_{\mathrm{SAMPLE-CE}_i}(\theta) = -\log\left(\frac{\exp(\beta s_{iy_i})}{\sum_{j\in S}\exp(\beta s_{jy_j})}\right). \tag{2}$$

It has been observed that using a larger set of negative samples leads to better performance [16]. A simple approach is to uniformly sample this set from $\mathcal{D}$ (popularly referred to as negative sampling). However, when $m$ is large, negative sampling is typically inefficient since it is difficult to obtain high-scoring irrelevant documents, thereby, providing a poor approximation of the cross-entropy loss. In the following sections, we explore highly scalable solutions for optimizing cross-entropy loss.

Random sampling is often implemented using *in-batch* negative sampling [15, 22, 16]. However, this approach is not scalable because huge amount of accelerator memory is required to achieve a bigger pool of in-batch negatives. For example, BERT [9] based transformers are typically used in NLP tasks but a single pair of (query, document) BERT-base consumes 600MB of accelerator memory during training. This further limits the effectiveness of random negative sampling.

## 2.3 Related Work

There have been many recent papers on using cached embeddings and nearest neighbors for improving negative mining during training. Several works consider caching embeddings for the entire dataset [13, 38]. They utilize a separate task to recompute embeddings based on the current parameters and recreate an index of nearest neighbor search. This can be computationally challenging for very large datasets, the cost of inference and index refresh scales linearly with the number of documents.

REALM [13] uses their embedding cache to optimize a softmax-cross entropy loss. They approximate the partition function by retrieving the $k$ largest scoring elements. This can be a poor approximation if the distribution is not highly concentrated around the top elements. Additionally, it requires $O(k)$ additional memory to perform backpropagation. ANCE [38] uses their cache to optimize a negative contrastive loss. MoCo [14] uses a streaming negative cache to increase the number of negative examples for feature embedding tasks. But it cannot be extended to dual encoder training, as the document encoder does not contribute to gradient updates.

Negative sampling is also widely used in classification problem with very large number of labels, where each document is considered a separate class. This setting is simpler than the dual encoder setting, since the model consists of one encoder and one large classification layer. The weights in the classification layer are only updated when their corresponding labels are included in a gradient step. In contrast, in the dual encoder setting, all document embeddings are changed during a gradient step.

For training with a softmax cross-entropy loss in the classification setting, Bengio and Senécal [4] develop an importance sampling approach for estimating the gradient of the cross-entropy loss function in large output spaces. Many following works utilize sampling techniques for estimating the gradient [17, 36, 30, 27, 25, 2]. Rawat et al. [30] analyzes the bias of different sampling techniques in gradient estimation. Mussmann et al. [28] use Gumbel-Max sampling to sample from the softmax distribution and accelerate gradient descent for certain problems. Zhang et al. [42] analyze the bias between gradients of the softmax loss and gradients generated from using hard negative mining. Other loss functions besides softmax cross-entropy include noise contrastive estimation, which was developed by [12] and utilized in [26, 25], ordered weighted losses [31], and triplet losses [33].

Hard and semi-hard negative mining is a collection of techniques for selecting negatives that the model is scoring above or near positive elements. Works in this area include [34, 37]. In particular, Dense Passage Retrieval [22] utilizes hard negatives generated by a BM25 ranker [32]. We note that hard negatives such as BM25 are often domain-specific and require finding a hard negative document for each query, which can be expensive with a large number of queries and documents.

We note that there are several approaches that take a dense retrieval model and use its retrieval results to improve performance further. The LTRe method [41] and the STAR and ADORE method [40] uses the retrieval results of ANCE to further improve performance with a ranking loss function. The RocketQA method [29] starts by training a model with sample negatives (in the cross-batch setting when training with multiple accelerators). This model is used to generate negatives for a cross-attention model, which is used to denoise the dataset to train an improved dense retrieval model. We compare to their Step1 model in the experimental section.

Our setting consists of learning from observed click data, thus we only see positive pairs. In the related *learning-to-rank* setting, the goal is to learn a ranking model using a set of positive and negative relevance labels. Works in this setting include [19, 1, 5, 21].

## 3 Full Document Cache

The primary objective is to find the parameters $\theta^*$ that minimize the softmax cross-entropy loss $\mathcal{L}_{\text{CE}}(\theta)$ (Equation 1). The standard algorithm to optimize the loss is stochastic gradient descent (or its variants). However, calculating stochastic gradients for this loss function is expensive in terms of both time and memory. Computation of the gradient involves: (1) a forward pass on *every* document in our dataset, (2) storing all intermediate activations (which are needed for the backward pass), and (3) calculating the gradients in the backward pass. Unfortunately, for large embedding encoder models such as those based on Transformers ([35]), this is prohibitively expensive. For instance, for a large BERT model on 1 million documents, the required memory would be hundreds of terabytes.

We propose an approach that can approximate the gradient of cross-entropy loss without needing to embed every document, thereby reducing the per-iteration computation and memory requirements for gradient computation. Our approach is based on cached embeddings [13, 38]. To understand our approach, we first start with gradient estimation with Gumbel-Max sampling. The following known fact shows that if we can sample from the softmax distribution then we can get an unbiased estimate of the stochastic gradient.

**Fact 1.** *The gradient of the cross-entropy has the following form:*

$$\nabla \mathcal{L}_{\text{CE}i} = -\beta \nabla s_{iy_i} + \sum_{j=1}^{m} p_{ij} \beta \nabla s_{ij},$$

*where $\mathcal{L}_{\text{CE}i} = -\log\left(\frac{\exp(\beta s_{iy_i})}{\sum_{j=1}^{m} \exp(\beta s_{ij})}\right)$ and $p_{ij} = \frac{\exp(\beta s_{ij})}{\sum_{j'=1}^{m} \exp(\beta s_{ij'})}$.*

*If $J$ is a sample from $p_i = (p_{i1}, p_{i2}, \ldots, p_{im})$, we have $\nabla \mathcal{L}_{\text{CE}i} = -\beta \nabla s_{iy_i} + \beta \text{E}[\nabla s_{iJ}]$.*

*Further, if $N_1, N_2, \ldots, N_m$ are i.i.d. standard Gumbel random variables, then the index $\arg\max_{j \in [m]} \beta s_{ij} + N_j$ is a sample from the distribution $p_i = (p_{i1}, p_{i2}, \ldots, p_{im})$.*

We refer to this approach as Gumbel-Max sampling and $\text{GumbelMaxSample}(\beta s)$ is used to represent this sampling procedure. A natural estimator of the gradient is $\nabla \widehat{\mathcal{L}}_{\text{CE}i} = -\beta \nabla s_{iy_i} + \beta \nabla s_{iJ}$ where $J$ is the index in $[m]$ obtained through Gumbel-Max sampling. Due to the feasibility of fast maximum inner product search (MIPS), Gumbel-Max sampling is an efficient way to sample from the softmax distribution; this, thereby, provides an efficient way to obtain an unbiased estimate of the gradient. In particular, the per-iteration memory requirement is reduced from $O(m\tau)$ to $O(md)$ where $\tau$ and $d$ are the model and embedding sizes respectively. This is due to the fact that the intermediate activations need to be stored only for documents $y_i$ and $J$, the negative obtained through Gumbel-Max sampling. When $d \ll \tau$ (which is typically the case while using large transformer models), this leads to a significant reduction in memory requirements. While the memory requirements are reduced drastically, we still need to do a forward pass on all documents to compute the scores $s_{ij}$. Using Fact 1 , we would need to embed every document every iteration, which is still computationally intensive.

**Algorithm 1** Cached Gumbel-Max Gradient Descent

---

Input: Learning rate $\eta$, Document refresh fraction $\rho \in (0, 1]$
Initialize parameters $\theta_0$.
Initialize embeddings table $\mathcal{E}$: $\mathcal{E}_j \leftarrow \phi_D(z_j; \theta_0)$ for all $j \in [m]$
**for** $t \in 0, 1, \ldots, T - 1$ **do**
    Sample $q_i, y_i$ from the training set
    $e_{q_i} \leftarrow \phi_Q(q_i; \theta_t)$
    $e_{z_{y_i}} \leftarrow \phi_D(z_{y_i}; \theta_t)$ and update $\mathcal{E}_{y_i} = e_{z_{y_i}}$
    Calculate scores $\tilde{s}_{ij} \leftarrow e_{q_i} \cdot \mathcal{E}_j$ for all $j \in [m]$
    $J \leftarrow \text{GumbelMaxSample}(\beta \tilde{s}_i)$
    $e_{z_J} \leftarrow \phi_D(z_J; \theta_t)$
    $s_{iJ} \leftarrow e_{q_i} \cdot e_{z_J}$
    $g_t \leftarrow -\beta \nabla s_{iy_i} + \beta \nabla s_{iJ}$
    $\theta_{t+1} \leftarrow \theta_t - \eta g_t$
    Select oldest $\rho m$ embeddings of $\mathcal{E}$ and update them to $\mathcal{E}_j \leftarrow \phi_D(z_j; \theta_{t+1})$
**end for**

---

This motivates an algorithm where we cache previously calculated embeddings to efficiently approximate the softmax distribution $p_i$. We first note the structure of the $s_{ij} = \phi_Q(q_i; \theta) \cdot \phi_D(z_j; \theta)$. The key computational challenge in Gumbel-Max sampling is computation of the embeddings $\phi_D(z_j; \theta)$ for all $j \in [m]$. This is required to find the index $J$ in Fact 1, which essentially renders it computationally intractable for large $m$. Our approach is to compute embeddings $\phi_D(z_j; \theta)$ for only a few documents $j \in [m]$ at each iteration and reuse the previously computed embeddings for the rest of the documents. In particular, let $z_j = \phi_D(z_j; \theta)$ be the current embedding of a document and $\tilde{z}_j = \phi_D(z_j; \tilde{\theta})$ be its previous embedding. If $\theta$ and $\tilde{\theta}$ are reasonably close, then $z_j$ and $\tilde{z}_j$ would be similar. In such a scenario, the scores $s_{ij}$ and $\tilde{s}_{ij}$, and, thereby the corresponding distributions $p_i = (p_{i1}, p_{i2}, \ldots, p_{im})$ and $\tilde{p}_i = (\tilde{p}_{i1}, \tilde{p}_{i2}, \ldots, \tilde{p}_{im})$ are also similar. As a consequence, the true gradient $\nabla \mathcal{L}_{\text{CE}i} = -\beta \nabla s_{iy_i} + \sum_{j=1}^{m} p_{ij} \beta \nabla s_{ij}$ and the approximation to the gradient $-\beta \nabla s_{iy_i} + \sum_{j=1}^{m} \tilde{p}_{ij} \beta \nabla s_{ij}$ should be similar. Note that for the approximation we take the gradient of $s_{ij}$ and not $\tilde{s}_{ij}$; we only replace the weight terms $p_{ij}$ with $\tilde{p}_{ij}$.

We described this algorithm in Algorithm 1. We present the algorithm with a batch size of 1—it can be easily extended to larger batches. We maintain an embedding table of all documents. At each iteration, we update only a small fraction $\rho$ of the stale embeddings to ensure that $p_i$ and $\tilde{p}_i$ are similar. Using this approach, we can approximately sample from this distribution using Gumbel-Max sampling in an efficient manner.

### 3.1 Theoretical Results

In this section, we establish error guarantees on the error of our Cached Gumbel-Max gradient approximation. Before delving into the technical details, we state the following key assumptions on the query and document encoder.

**Assumption 1.** *The following conditions hold for the query $\phi_Q$ and document encoder $\phi_D$:*

    *A1 The query and document encoder functions are both $L$-Lipschitz in the parameters $\theta$. In particular, we have $\|\phi(q_i; \theta) - \phi(q_i; \theta')\|_2 \leq L\|\theta - \theta'\|_2$ for all $i \in [n]$ and $\|\phi(z_i; \theta) - \phi(z_i; \theta')\|_2 \leq L\|\theta - \theta'\|_2$ for all $i \in [m]$.*

    *A2 The query and document embeddings are bounded i.e., we have $e_q, e_z = \phi_Q(q; \theta), \phi_D(z; \theta)$ satisfy $\|e_q\|_2, \|e_z\|_2 \leq 1$.*

    *A3 The score functions have bounded gradients i.e., we have $\|\nabla s_{ij}\|_2 \leq M$ for all $i \in [n]$ and $j \in [m]$.*

All of these assumptions are fairly mild and are common in optimization literature. As noted earlier, the second assumption does not lead to much loss of generality as a larger bound on the norm can be absorbed into the inverse temperature parameter $\beta$. In the following result, we first show that the error can be bounded by the $\ell_\infty$ error between the true scores and the scores with the cached embeddings.

We then show that if the cached embeddings are generated from Algorithm 1 with learning rate $\eta$ and refresh rate $\rho$, we can bound the gradient error at each iteration.

**Theorem 2.** *Let $\theta_t$ and index $i$ be the parameters and training point selected at $t^{th}$ iteration of Algorithm 1, respectively. Let $J$ represent the Gumbel-Max index selected at that iteration. Then, under Assumption 1, we have the following gradient approximation with the cached embeddings*

$$\nabla\tilde{\mathcal{L}}_{\mathrm{CE}_i}(\theta_t) = \mathbb{E}_J[g_t] = -\beta\nabla s_{iy_i} + \sum_{j=1}^{m} \tilde{p}_{ij}\beta\nabla s_{ij},$$

*where $\tilde{p}_{ij} = \frac{\exp(\beta\tilde{s}_{ij})}{\sum_{j'=1}^{m}\exp(\beta\tilde{s}_{ij'})}$ and $\tilde{s}_{ij} = e_{q_i}\cdot\mathcal{E}_j$. Furthermore, we have the following bound on the gradient approximation of $\nabla\mathcal{L}_{\mathrm{CE}i}(\theta_t)$:*

$$\|\nabla\mathcal{L}_{\mathrm{CE}i}(\theta_t) - \nabla\tilde{\mathcal{L}}_{\mathrm{CE}_i}(\theta_t)\|_2 \leq 2\beta^2 M\|\tilde{s}_i - s_i\|_\infty.$$

*When parameter updates are generated by Algorithm 1 with step size $\eta$ and update rate $\rho$, we have*

$$\|\nabla\mathcal{L}_{\mathrm{CE}i}(\theta_t) - \nabla\tilde{\mathcal{L}}_{\mathrm{CE}_i}(\theta_t)\|_2 \leq 4\eta\beta^3 LM^2\left(\frac{1}{\rho} - 1\right).$$

We see that the error can be controlled by either increasing the refresh rate or decreasing the gradient norm. This provides a bound on the bias in the gradient approximation. Using the above result, we show the following first-order convergence guarantees. For proving convergence, we need the following additional assumption.

**Assumption 2.** *We assume loss function $\nabla\mathcal{L}_{\mathrm{CE}_i}$ is S-smooth i.e., we have $\|\nabla\mathcal{L}_{\mathrm{CE}_i}(\theta) - \nabla\mathcal{L}_{\mathrm{CE}_i}(\theta')\|_2 \leq S\|\theta - \theta'\|_2$ holds for all $\theta, \theta' \in \mathbb{R}^p$ and $i \in [n]$.*

Under the above assumption, we have the following convergence result in general non-convex settings.

**Theorem 3.** *Suppose we run Algorithm 1 for $T$ iterations with stepsize $\eta = \frac{\sqrt{\mathcal{L}_{\mathrm{CE}}(\theta_0) - \mathcal{L}_{\mathrm{CE}}(\theta^*)}}{\sqrt{2TSM}}$.*

*Then under Assumption 1 and 2, we have the following:*

$$\frac{1}{T}\sum_{t=0}^{T}\mathbb{E}[\|\nabla\mathcal{L}_{\mathrm{CE}}(\theta_t)\|_2^2] \leq 4M\sqrt{\frac{S(\mathcal{L}_{\mathrm{CE}}(\theta_0) - \mathcal{L}_{\mathrm{CE}}(\theta^*))}{T}} + \frac{4\beta^6 L^2 M^2(\mathcal{L}_{\mathrm{CE}}(\theta_0) - \mathcal{L}_{\mathrm{CE}}(\theta^*))}{ST}\left(\frac{1}{\rho} - 1\right)^2.$$

We have that $-2\beta + \log m \leq \mathcal{L}_{\mathrm{CE}}(\theta) \leq 2\beta + \log m$. Thus the term $\mathcal{L}_{\mathrm{CE}}(\theta_0) - \mathcal{L}_{\mathrm{CE}}(\theta^*) \leq 4\beta$.

We observe that the bias introduced due to stale embeddings is a lower order term in the bound of Theorem 3. In particular, one can use $\rho = \frac{1}{1+T^{1/4}}$ without affecting the convergence rate of the standard SGD algorithm. For a large $T$ (which is typical in machine learning settings), this can have a significant impact on the computational complexity since a very small fraction of the documents need to be updated at each iteration.

## 3.2 Computational Discussion

We store our cache using accelerator memory. This prevents the need to have a separate task that constantly reindexes the embeddings as they change throughout training. This is feasible for moderate size datasets. For instance, 1 million training points with an embedding dimension of 512 and feature vector dimension of 1024 will use about 6 GB memory, which can fit on a single accelerator. To calculate the perturbed nearest neighbor for Gumbel-Max sampling, we simply brute force calculate the largest dot product. Accelerators such as GPUs and TPUs can do this very efficiently—in our experiments we see that the steps/second increases only a small amount as we increase the cache size. This is because the cost of computing the embeddings is much larger than the cost of doing the search due to the complexity of large transformer models.

If faster sampling is needed, it can be accelerated by using fast nearest neighbor search. Mussman et al. [28] consider an approach that applies Gumbel perturbations to the $k$ highest scoring elements plus a small number of random elements to perform Gumbel-Max sampling in sublinear time.

However, having the cache on an accelerator has limitations due to the accelerator memory limits. In the next section, we modify Algorithm 1 to be a streaming algorithm.

### 3.3 Conditional Sampling Negatives

Note that Gumbel-Max sampling has a chance to sample the positive element. If this happens, then our gradient approximation is zero. Since we are training our model to make the score of the positive element large, this can happen often. We develop an approach to force the sampled element to be a negative element while maintaining a similar expected gradient.

Assume that the positive element is $z_1$ and the negative elements are $z_2, z_3, \ldots, z_m$. We have that

$$\nabla \mathcal{L}_{\mathrm{CE}i} = -\beta \nabla s_{i1} + \sum_{j=1}^{m} p_{ij} \beta \nabla s_{ij} = -\beta(1 - p_{i1}) \nabla s_{i1} + (1 - p_{i1}) \sum_{j=2}^{m} \frac{p_{ij}}{1 - p_{i1}} \beta \nabla s_{ij}.$$

Note that $\left( \frac{p_{i2}}{1-p_{i1}}, \frac{p_{i3}}{1-p_{i1}}, \cdots, \frac{p_{im}}{1-p_{i1}} \right)$ is the conditional distribution where we condition on not sampling the first element. Thus we can sample from the conditional distribution as long as we properly scale the gradient by $1 - p_{i1}$. We note that the value of $p_{i1}$ depends on the negative embeddings. Since we have all the negatives on our accelerator, we can also calculate the partition function without much additional compute, as it has the same complexity as of our nearest neighbor search.

## 4 Streaming Cache

For very large datasets, the Gumbel-Max sampling step can become difficult. If the embedding vectors are stored in accelerator memory, then only a finite number of vectors can be stored on a fixed computational budget. If the vectors are stored in CPU memory with a CPU-based retrieval system, then the nearest neighbor index needs to be constantly recreated, which is computationally expensive for large datasets. Additionally, we see in Theorem 2 that as the fraction of elements refreshed each iteration $\rho$ decreases, then the learning rate needs to decrease as well to maintain a given bias in gradient estimation. Alternatively, more embeddings will need to be refreshed each iteration as the size of the dataset grows.

To address this issue, we develop a streaming variant of our algorithm. The key difference with respect to the full dataset setting is that instead of storing all the document embeddings in memory, we only store a sample multiset $\mathcal{S}$ of size $\alpha m$. After every iteration we remove the oldest $\rho \alpha m$ elements in $\mathcal{S}$ and replace them with $\rho \alpha m$ new elements sampled uniform i.i.d. from the dataset. The fraction $\alpha$ can be tuned to fit a given computational budget.

Our gradient estimator approximates gradients from a *cache cross-entropy* loss. We define the cache cross-entropy loss to be the following:

$$\mathcal{L}_{\mathrm{CacheCE}i} = -\log \frac{\exp(\beta s_{y_i})}{\exp(\beta s_{iy_i}) + \frac{1}{\alpha} \sum_{j \in \mathcal{S}, j \neq y_i} \exp(\beta s_{ij})}$$

Following [4], we scale the weight of the negative elements by $\frac{1}{\alpha}$, otherwise the partition function would underestimate the true partition function. Thus is equivalent to shifting the scores of the negative elements by $\frac{1}{\beta} \log \frac{1}{\alpha}$.

The use of the cache cross-entropy loss $\mathcal{L}_{\mathrm{CacheCE}}$ instead of the true cross entropy loss induces a bias in gradient estimation. We are able to show that the bias scales inversely with the cache size. The proof of Lemma 4 is a modification of the proof of Theorem 1 in [30].

**Lemma 4.** *Assume Assumptions A2 and A3 hold, i.e., that the norm of the encoder embeddings is bounded by $1$ and the norm of the gradients of the scores is bounded by $M$.*

*We have that*

$$\| \nabla \mathcal{L}_{\mathrm{CE}i} - \mathrm{E}\left[ \nabla \mathcal{L}_{\mathrm{CacheCE}i} \right] \|_2 \leq \frac{\exp(O(\beta)) M}{\alpha m},$$

*where the expectation is taken over the randomness of the elements in the cache.*

In Algorithm 2 we present our Algorithm for training with the streaming cache. We use $\mathrm{GumbelMaxSample}(\beta s, \mathcal{S})$ to denote the Gumbel-Max sampling described in Fact 1 with scores restricted to documents in set $\mathcal{S}$.

There are two sources of bias in our gradient estimate: first due to the staleness of the cache and the second due to using a sampled set of negatives rather than the entire dataset. We handle the latter in

**Algorithm 2** Streaming Cached Gumbel-Max Gradient Descent

---

Input: Learning rate $\eta$, Cache fraction $\alpha \in (0, 1]$, Document refresh fraction $\rho \in (0, 1]$
Initializes parameters $\theta_0$.
Random sample $\mathcal{S} \subseteq [m]^{\alpha m}$.
Initialize embeddings table $\mathcal{E}$: $\mathcal{E}_j \leftarrow \phi_D(z_j; \theta_0)$ for all $j \in \mathcal{S}$
**for** $t \in 0, 1, \ldots, T - 1$ **do**
    Sample $q_i, y_i$ from the training set
    $e_{q_i} \leftarrow \phi_Q(q_i; \theta_t)$
    $e_{z_{y_i}} \leftarrow \phi_D(z_{y_i}; \theta_t)$ and update $\mathcal{E}_{y_i} = e_{z_{y_i}}$
    Calculate scores $\tilde{s}_{ij} \leftarrow e_{q_i} \cdot \mathcal{E}_j + \log(1/\alpha)/\beta$ for all $j \in \mathcal{S}$ and $\tilde{s}_{iy_i} = e_{q_i} \cdot e_{z_{y_i}}$
    $\hat{\mathcal{S}} \leftarrow \mathcal{S}$ with all instances of $y_i$ removed
    $J \leftarrow \text{GumbelMaxSample}(\beta \tilde{s}_i, \hat{\mathcal{S}})$
    $e_{z_J} \leftarrow \phi_D(z_J; \theta_t)$
    $s_{iJ} \leftarrow e_{q_i} \cdot e_{z_J}$
    $p_{y_i} \leftarrow$ probability of $y_i$ under $\text{softmax}(\tilde{s}_i)$
    $g_t \leftarrow \beta(1 - p_{y_i})(-\nabla s_{iy_i} + \nabla s_{iJ})$
    $\theta_{t+1} \leftarrow \theta_t - \eta g_t$
    Select oldest $\rho \alpha m$ embeddings set $O$ of $\mathcal{E}$ and remove them from $\mathcal{E}$
    Sample $\mathcal{S}' \subseteq [m]$, $|\mathcal{S}'| = \rho \alpha m$
    Update $\mathcal{E}_j \leftarrow \phi_D(z_j; \theta_{t+1})$ for $j \in \mathcal{S}'$
    Update set $\mathcal{S} \leftarrow \mathcal{S} - O \cup \mathcal{S}'$
**end for**

---

Lemma 4. For the former, note that our error guarantees in Theorem 2 still apply to the streaming cache, as we can bound the error between true gradient using this set of negatives with our gradient approximation. This allows us to bound the bias due to the staleness of the cache.

Since we were able to control the bias of our gradient estimator even with the streaming cache, we can establish a first order convergence theorem for training with Algorithm 2.

**Theorem 5.** *Assume that Assumptions 1 and 2 hold and we use a learning rate* $\eta = \frac{\sqrt{\mathcal{L}_{\text{CE}}(\theta_0) - \mathcal{L}_{\text{CE}}(\theta^*)}}{\sqrt{2TSM}}$. *Running Algorithm 2 with a cache of size* $\alpha m$ *with a refresh rate* $\rho$ *creates updates* $\theta_1, \theta_2, \ldots, \theta_T$ *such that*

$$\frac{1}{T} \sum_{t=0}^{T} \mathbb{E}[\|\nabla \mathcal{L}_{\text{CE}}(\theta_t)\|_2^2] \leq O\left(\frac{1}{\sqrt{T}} + \frac{1}{\alpha^2 m^2} + \frac{1}{T}\left(\frac{1}{\rho} - 1\right)^2\right),$$

*where we omit terms that depend on S, L, M, and* $\beta$.

If the cache size satisfies $\alpha m \geq T^{1/4}$, then we do not asymptotically affect the rate of convergence.

We see that using a streaming cache smaller than the entire dataset adds only a small amount of bias in convergence and reduces the computational burden of storing embeddings and calculating nearest neighbors significantly.

## 5 Experiments

We analyse the performance of our approach on the MS MARCO passage retrieval task [3] and the TREC 2019 passage retrieval task [8]. Both tasks utilize the same dataset consisting of 500,000 (query, passage) pairs and 8.8 million passages in the database. MS MARCO measures the mean reciprocal rank at 10(MRR@10) with a holdout set of query, passage pairs. TREC 2019 measures the normalized discounted cumulative gains at 10 (NDCG@10) compared to a set of human labeled relevance scores.

| Method | Batch Size | Negatives | Updates/Step | Memory/Step | Steps/Sec |
|---|---|---|---|---|---|
| Sample Cross-Entropy | 8 | 16 | - | 10G | 9.95 |
| Sample Cross-Entropy | 8 | 64 | - | 25G | 4.64 |
| Sample Cross-Entropy | 8 | 256 | - | 87G (OOM) | 1.43 (est.) |
| Cached Negative | 8 | 8k | 256 | 15G | 3.38 |
| Cached Negative | 8 | 32k | 256 | 15G | 3.38 |
| Cached Negative | 8 | 128k | 256 | 16G | 3.37 |
| Cached Negative | 8 | 512k | 256 | 19G | 3.32 |
| Cached Negative | 8 | 2M | 512 | 38G | 2.04 |
| Full Dataset Cache | 8 | 8.8M | 512 | 90G (OOM) | - |

Table 1: Runtime and resource consumption for various configurations of cache sizes and number of random negatives. Since sample cross-entropy doesn't fit in memory we estimate the steps/second with linear interpolation.

## 5.1 MSMARCO / TREC Experimental Set Up

Following related work [22, 38], we start with a pretrained BERT-base model [9] [1]. Both our query and document encoder are initialized with this model and share parameters during training. We add a fully connected layer to the end of the encoder that projects to 512 dimensions. We do no additional pre-training. We use a global batch size of 8, the Adam optimizer [23], and we train for $250,000$ steps. We normalize the output embeddings and have a trainable parameter $\beta$ scale the scores. We use a learning rate of $1 \times 10^{-5}$ for all experiments except when we train with a cache of 2 million elements, where we use a learning rate of $5 \times 10^{-5}$. This is due to the increased staleness of the large cache. Our experiments use 8 V2 Cloud TPUs. Each replica on the TPU has 8GB memory, for a total of 64GB memory.

## 5.2 MSMARCO / TREC Computational Performance

We first examine the computational performance of our method. Since we use brute force to calculate the negative with the highest perturbed score, we are interested in seeing how this affects the runtime performance of our algorithm. Randomly sampled passages are used as negatives for sample cross-entropy and the updates for the negative cache.

Due to the increased staleness of large caches, we do 512 updates when using a cache with 2 million elements. For all other cache sizes we use 256 elements.

We see in Table 1 that the cache size can scale to hundreds of thousands of elements with minimal effect on the speed of training. The cost of calculating embeddings and gradients with large transformer models is significantly more expensive than the cost of the nearest neighbor search, even if we use a brute force approach.

Further, we see that using a cache with 2 million elements requires less memory than using just 256 sample negatives, even when 512 elements are updated every iteration. This is because the memory required for the backward step is about 10x larger than the memory needed for the forward step.

Additionally, we see that there is a limit to the cache size. If we were to use all 8.8M negatives, we would not be able to fit them on 8 V2 Cloud TPUs and would have to move to a separate nearest neighbor system. Our streaming approach allows us to adapt the cache size to the available memory.

## 5.3 MSMARCO / TREC Experimental Results

We now analyze the statistical improvement of using a large negative cache. In Table 2 we present the output MRR@10 on the development set of MS MARCO passage retrieval task and NDCG@10 on the TREC 2019 passage retrieval task for training with stochastic negative mining, our negative cache approach, as well as two external benchmarks, ANCE [38] and DPR [22].

We see that our streaming cache method achieves similar statistical performance to ANCE without needing to store and search all elements in the dataset. Additionally we see that it outperforms all

---

[1]Model obtained from `https://tfhub.dev/tensorflow/bert_en_uncased_L-12_H-768_A-12/4`.

| Method | MS MARCO (MRR@10) | TREC (NDCG@10) |
|---|---|---|
| Sample Cross-Entropy (16 negatives) | 0.235 | 0.493 |
| Sample Cross-Entropy (64 negatives) | 0.257 | 0.513 |
| Sample Cross-Entropy (256 negatives) | 0.273 | 0.538 |
| Negative Cache (8k negatives) | 0.310 | 0.600 |
| Negative Cache (32k negatives) | 0.315 | 0.630 |
| Negative Cache (128k negatives) | 0.323 | 0.633 |
| Negative Cache (512k negatives) | 0.322 | 0.649 |
| Negative Cache (2M negatives) | 0.331 | 0.630 |
| ANCE (NCE with 8M negatives) [38] | 0.330 | 0.648 |
| DPR (Random + BM25 Negative) [22] | 0.311 | 0.600 |
| RocketQA Step1 BERT-base [29] | 0.327 | - |

Table 2: Statistical performance of stochastic negative mining, negative cache training, and related dual encoder training benchmarks for the MSMARCO and TREC task. Values for DPR were obtained from [38].

| Method | Recall@20 | Recall@100 |
|---|---|---|
| Cache (2M negatives) | 0.784 | 0.856 |
| ANCE (NCE with 20M negatives) [38] | 0.819 | 0.875 |
| DPR (Random + BM25 Negative) [22] | 0.784 | 0.854 |
| RocketQA Step1 BERT-base [29] | - | 0.860 |

Table 3: Statistical performance of negative cache training and related dual encoder training benchmarks for the Natural Question task.

sample cross-entropy approaches with a smaller memory footprint. Further, we see that our approach outperform the BM25 negatives utilized by DPR.

### 5.4   Natural Question Experimental Results

We now experimentally compare our method on the Natural Question dataset [24]. We follow the methodology used by [22]. The goal of the experiment is to learn a model that matches questions with passages that contain the answer to the question. The dataset consists of about 60,000 query, positive passage pairs and there are about 20 million passages. In evaluation, we consider a retrieved passage a positive if it contains an exact match to one of the labeled answers to the given question.

In Table 3 we compare training with a negative cache with 2 million negatives to several state-of-the-art dual encoder training methods. We see that training with a negative cache compares favorably with DPR without the use of a domain-specific hard negative mining techniques such as BM25. The use of such a techniques requires an inference step to find the hard negative among all passages, which can be expensive for large datasets. Additionally we note that ANCE is initialized with the DPR model and we use only 10% of the negatives in our cache.

## 6   Conclusion

In this paper we developed an algorithm for efficiently estimating gradients for dual encoder models with a softmax cross-entropy loss function. We see that we can efficiently obtain gradient estimations with low bias with an appropriately chosen learning rate and a sufficiently large negative cache. We analyze first order convergence of training with a streaming negative cache and establish near-optimal convergence bounds compared to SGD for reasonable parameter choices. In dense retrieval experiments we see that our approach is efficient and can compare favorably to state-of-the-art dual encoder training methods.

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
