# Appendix

## 7 Proof of Theorem 2

*Proof.* Let $e_{q_i} = \phi_Q(q_i; \theta_t)$ be the query embedding, $e_{z_i} = \phi_D(z_i; \theta_t)$ be the document embedding, and $e_{z_J} = \phi_D(z_J; \theta_t)$. Recall that $\mathcal{E}_1, \mathcal{E}_2, \ldots, \mathcal{E}_m$ be the (potentially stale) embeddings in the cache. Let $s^+ = e_{q_i} \cdot e_{z_i}$, $s_j = e_{q_i} \cdot \phi_D(z_j; \theta_t)$, $\tilde{s}_j = e_{q_i} \cdot \mathcal{E}_j$.

Recall that

$$\mathcal{L}_{CE_i}(\theta_t) = -\log\left( \frac{\exp(\beta s^+)}{\exp(\beta s^+) + \sum_{j \neq y_i} \exp(\beta s_j)} \right).$$

For simplicity, we use $\nabla$ and $\tilde{\nabla}$ to denote $\nabla \mathcal{L}_{CE_i}(\theta_t)$ and $\nabla \tilde{\mathcal{L}}_{CE_i}(\theta_t)$ respectively. We first observe that

$$\tilde{\nabla} = \mathbb{E}_J[g_t] = -\beta \nabla s^+ + \sum_j \tilde{p}_j \beta \nabla s_j.$$

This follows as simple consequence of the Gumbel-Max sampling. Furthermore, we have

$$\nabla = -\beta \nabla s^+ + \sum_j p_j \beta \nabla s_j.$$

From the above expression, we have that

$$\|\nabla - \tilde{\nabla}\|_2 = \beta \| \sum_j (p_j - \tilde{p}_j) \nabla s_j \|_2$$

$$\leq \beta \sum_j |p_j - \tilde{p}_j| \|\nabla s_j\|_2$$

$$\leq \beta M \|p - \tilde{p}\|_1.$$

The last inequality follows from bounded nature of the score $\|\nabla s_j\| \leq M$. Consider a term $p_j - \tilde{p}_j$. We have that

$$p_j - \tilde{p}_j = \frac{\exp(\beta s_j)}{\sum_l \exp(\beta s_l)} - \frac{\exp(\beta \tilde{s}_j)}{\sum_l \exp(\beta \tilde{s}_l)}$$

$$= \frac{\exp(\beta s_j)}{\sum_l \exp(\beta s_l)} - \frac{\exp(\beta(s_j + (\tilde{s}_j - s_j)))}{\sum_l \exp(\beta(s_l + (\tilde{s}_l - s_l)))}$$

$$\leq \frac{\exp(\beta s_j)}{\sum_l \exp(\beta s_l)}(1 - \exp(-\beta\|\tilde{s} - s\|_\infty))$$

$$= p_j(1 - \exp(-2\beta\|\tilde{s} - s\|_\infty))$$

$$\leq 2p_j\beta\|\tilde{s} - s\|_\infty$$

Similarly, we have that

$$\tilde{p}_j - p_j \leq 2\tilde{p}_j\beta\|\tilde{s} - s\|_\infty.$$

Thus we have that $|p_i - \tilde{p}_i| \leq 2\beta\|\tilde{s} - s\|_\infty(p_i + \tilde{p}_i)$ and thus

$$\|p - \tilde{p}\|_1 \leq 4\beta\|\tilde{s} - s\|_\infty$$

We bound $\|\tilde{s} - s\|_\infty$ as follows. Suppose it is at most $k$ updates since any embedding in $\mathcal{E}$ has been updated. In particular, let $t_j$ denote the time step when $j$ was last updated in $\mathcal{E}$. Then, we have

$$|\tilde{s}_j - s_j| = \left| e_{q_i} \cdot \mathcal{E}_j - e_{q_i} \cdot e_{z_j} \right|$$

$$\leq \|e_{q_i}\|_2 \left\| \mathcal{E}_j - e_{z_j} \right\|_2$$

$$\leq \left\| \mathcal{E}_j - e_{z_j} \right\|_2$$

$$= \left\| \phi_D(z_j; \theta_t) - \phi_D(z_j; \theta_{t_j}) \right\|$$

$$\leq L \left\| \theta_t - \theta_{t_j} \right\| \leq \eta\beta LM(t - t_j)$$

Thus we have that $\|\nabla - \tilde{\nabla}\|_2 \leq 4\eta\beta^3 LM^2 k$. When using the refresh fraction of $\rho$, it can be shown the $k$ is in expectation of the order $\frac{1}{\rho} - 1$, which completes the proof. $\square$

# 8 Proof of Theorem 3

To prove Theorem 3, we start with the following result.

**Lemma 6.** *Let $\mathcal{L} = \frac{1}{m}\sum_{i=1}^{m}\mathcal{L}_{\mathrm{CE}_i}$. Assume that a loss function $\mathcal{L}_{CE_i}(\theta)$ satisfies:*

- *(Bounded Gradients) We have that $\|\mathcal{L}_{\mathrm{CE}_i}(\theta)\| \le 2M$ for all parameters $\theta \in \mathbb{R}^p$.*

- *(Smoothness) We have that $\|\nabla\mathcal{L}_{\mathrm{CE_i}}(\theta) - \nabla\mathcal{L}_{\mathrm{CE_i}}(\theta')\|_2 \le S\|\theta - \theta'\|_2$.*

*Furthermore, suppose we run an approximate stochastic gradient descent with stochastic gradient with bounded bias, $\|\mathbb{E}[g_t \mid \theta_t] - \nabla\mathcal{L}(\theta_t)\|_2 \le \Delta_t$, and additionally $\|g_t\| \le M$ for all $t \in [T]$. If we update our parameters with a stepsize $\eta$, we have that*

$$\frac{1}{T}\sum_{t=0}^{T}\mathbb{E}[\|\nabla\mathcal{L}(\theta_t)\|_2^2] \le \frac{\mathcal{L}(\theta_0) - \mathcal{L}(\theta^*)}{\eta T} + \frac{1}{2T}\sum_{t=0}^{T}\Delta_t^2 + 2\eta S M^2.$$

*Proof.* From the Lipschitz continuous nature of the function $\mathcal{L}$, we have

$$\mathbb{E}[\mathcal{L}(\theta_{t+1})] \le \mathbb{E}\left[\mathcal{L}(\theta_t) + \nabla\mathcal{L}(\theta_t)\cdot(\theta_{t+1} - \theta_t) + \frac{S}{2}\|\theta_{t+1} - \theta_t\|_2^2\right]$$

$$= \mathbb{E}\left[\mathcal{L}(\theta_t) - \eta\nabla\mathcal{L}(\theta_t)\cdot g_t + \frac{\eta^2 S}{2}\|g_t\|_2^2\right]$$

$$\le \mathbb{E}\left[\mathcal{L}(\theta_t) - \eta\|\nabla\mathcal{L}(\theta_t)\|_2^2 - \eta\nabla\mathcal{L}(\theta_t)\cdot\Delta_t\right] + \frac{4\eta^2 S M^2}{2}$$

$$\le \mathbb{E}\left[\mathcal{L}(\theta_t) - \eta\|\nabla\mathcal{L}(\theta_t)\|_2^2 + \eta\Delta_t\|\nabla\mathcal{L}(\theta_t)\|_2\right] + 2\eta^2 S M^2.$$

The second inequality follows from bounded nature of $g_t$. The above inequality can be further bounded in the following manner:

$$\mathbb{E}[\mathcal{L}(\theta_{t+1})] \le \mathbb{E}\left[\mathcal{L}(\theta_t) - \eta\|\nabla\mathcal{L}(\theta_t)\|_2^2 + \eta\Delta_t\|\nabla\mathcal{L}(\theta_t)\|_2\right] + 2\eta^2 S M^2$$

$$\le \mathbb{E}[\mathcal{L}(\theta_t)] - \frac{\eta}{2}\mathbb{E}[\|\nabla\mathcal{L}(\theta_t)\|_2^2] + \frac{\eta}{2}\Delta_t^2 + 2\eta^2 S M^2.$$

The second inequality follows from the fact that $ab \le (a^2 + b^2)/2$. Summing over all $t \in [0, T]$ and using telescoping sum, we have

$$\frac{1}{T}\sum_{t=0}^{T}\mathbb{E}[\|\nabla\mathcal{L}(\theta_t)\|_2^2] \le \frac{\mathcal{L}(\theta_0) - \mathcal{L}(\theta_T)}{\eta T} + \frac{1}{2T}\sum_{t=0}^{T}\Delta_t^2 + 2\eta S M^2. \tag{3}$$

This completes the proof of the lemma.

$\square$

We now focus on the proof of Theorem 3..

*Proof.* We first note that under the assumptions of Theorem 3, $\|\nabla\mathcal{L}_{\mathrm{CE}}(\theta_t)\| \le 2M$ and $\|g_t\| \le 2M$. This simply follows from the structure of $\nabla\mathcal{L}_{\mathrm{CE}}$. Using the above lemma, we have the following:

$$\frac{1}{T}\sum_{t=0}^{T}\mathbb{E}[\|\nabla\mathcal{L}_{\mathrm{CE}}(\theta_t)\|_2^2] \le \frac{\mathcal{L}_{\mathrm{CE}}(\theta_0) - \mathcal{L}_{\mathrm{CE}}(\theta^*)}{\eta T} + 8\eta^2\beta^6 L^2 M^4\left(\frac{1}{\rho} - 1\right)^2 + 2\eta S M^2.$$

This follows simply from the bias bounded obtained in Theorem 2. Using $\eta = \frac{\sqrt{\mathcal{L}_{\mathrm{CE}}(\theta_0) - \mathcal{L}_{\mathrm{CE}}(\theta^*)}}{\sqrt{2TSM}}$ specified in the theorem, we obtain

$$\frac{1}{T}\sum_{t=0}^{T}\mathbb{E}[\|\nabla\mathcal{L}_{\mathrm{CE}}(\theta_t)\|_2^2] \le 4M\sqrt{\frac{S(\mathcal{L}_{\mathrm{CE}}(\theta_0) - \mathcal{L}_{\mathrm{CE}}(\theta^*))}{T}} + \frac{4\beta^6 L^2 M^2(\mathcal{L}_{\mathrm{CE}}(\theta_0) - \mathcal{L}_{\mathrm{CE}}(\theta^*))}{ST}\left(\frac{1}{\rho} - 1\right)^2.$$

This completes the proof of Theorem 3.

$\square$

# 9  Proof of Lemma 4 and Theorem 5

We use the following lemma in the proof of Lemma 4.

**Lemma 7** (Lemma 5 in [30]). *Given a random variable $V \geq a > 0$, we have that*

$$\frac{1}{E[V]} \leq E\left[\frac{1}{V}\right] \leq \frac{1}{E[V]} + \frac{\text{Var}(V)}{a^3}.$$

We now prove Lemma 4.

*Proof.* Our proof follows the proof approach in Theorem 1 in [30], modified to work with an $\ell_2$ bound on the score gradients and simplified for our sampling scheme.

Assume that the positive element is $z_1$ and thus the negative elements are $z_2, \ldots, z_m$.

Let $U = \exp(\beta s_1)\beta \nabla s_1 + \frac{1}{\alpha}\sum_{j \in \mathcal{S}}\exp(\beta s_j)\beta \nabla s_j$ and $V = \exp(\beta s^+) + \frac{1}{\alpha}\sum_{j \in \mathcal{S}}\exp(\beta s_j)$. We have that $-\beta \nabla s_1 + \frac{E[U]}{E[V]} = \nabla \mathcal{L}_{\mathcal{CE}i}$ and $E[g] = -\beta \nabla s_1 + E\left[\frac{U}{V}\right]$ We thus want to show that $E\left[\frac{U}{V}\right] \approx \frac{E[U]}{E[V]}$.

Let $k_1, k_2, \ldots, k_c$ be the $c$ elements of $\mathcal{S}$. We have that

$$E\left[\frac{U}{V}\right] = E\left[\frac{\exp(\beta s_1)\beta \nabla s_1 + \frac{1}{\alpha}\sum_{j=1}^{c}\exp(\beta s_{k_j})\beta \nabla s_{k_j}}{\exp(\beta s_1) + \frac{1}{\alpha}\sum_{j=1}^{c}\exp(\beta s_{k_j})}\right]$$

$$= \exp(\beta s_1)\beta \nabla s_1 E\left[\frac{1}{V}\right] + E\left[\frac{\frac{1}{\alpha}\sum_{j=1}^{c}\exp(\beta s_{k_j})\beta \nabla s_{k_j}}{\exp(\beta s_1) + \frac{1}{\alpha}\sum_{j=1}^{c}\exp(\beta s_{k_j})}\right] \quad (4)$$

We first bound the first term in Equation (4) from above and below.

We have that $V \geq m\exp(-\beta)$ and $\text{Var}(V) \leq c\frac{\exp(2\beta)}{\alpha^2}$. Thus by Lemma 7 we have that

$$\frac{1}{E[V]} \leq E\left[\frac{1}{V}\right] \leq \frac{1}{E[V]} + \frac{c\frac{\exp(2\beta)}{\alpha^2}}{m^3\exp(-3\beta)} = \frac{1}{E[V]} + \frac{\exp(5\beta)}{\alpha m^2}.$$

This implies that

$$\frac{\exp(\beta s_1)\beta \nabla s_1}{Z} \leq \exp(\beta s_1)\beta \nabla s_1 E\left[\frac{1}{V}\right] \leq \frac{\exp(\beta s_1)\beta \nabla s_1}{Z} + \frac{\exp(6\beta)\beta|\nabla s_1|}{\alpha m^2} \quad (5)$$

We now bound the second equation in Equation (4).

Let $S_{c-1} = \sum_{j=1}^{c-1}\exp(\beta s_{k_j})$. We have that

$$E\left[\frac{\frac{1}{\alpha}\sum_{j=1}^{c}\exp(\beta s_{k_j})\beta \nabla s_{k_j}}{\exp(\beta s_1) + \frac{1}{\alpha}\sum_{j=1}^{c}\exp(\beta s_{k_j})}\right] = \frac{c}{\alpha}E\left[\frac{\exp(\beta s_{k_c})\beta \nabla s_{k_c}}{\exp(\beta s_1) + \frac{1}{\alpha}S_{c-1} + \frac{1}{\alpha}\exp(\beta s_{k_c})}\right]$$

$$= \frac{c}{\alpha m}\sum_{i=2}^{m}\exp(\beta s_i)\beta \nabla s_i E\left[\frac{1}{\exp(\beta s_1) + \frac{1}{\alpha}S_{c-1} + \frac{1}{\alpha}\exp(\beta s_i)}\right]$$

$$= \sum_{i=2}^{m}\exp(\beta s_i)\beta \nabla s_i E\left[\frac{1}{\exp(\beta s_1) + \frac{1}{\alpha}S_{c-1} + \frac{1}{\alpha}\exp(\beta s_i)}\right] \quad (6)$$

Now we have that

$$E\left[\exp(\beta s_1) + \frac{1}{\alpha}S_{c-1} + \frac{1}{\alpha}\exp(\beta s_i)\right] = \exp(\beta s_1) + \frac{c-1}{c}Z^- + \frac{1}{\alpha}\exp(\beta s_i)$$

$$= Z - \frac{1}{c}Z^- + \frac{1}{\alpha}\exp(\beta s_i),$$

where $Z^-$ is the partition function restricted to just the negatives.

Using $Z \geq Z^-$ and $m \exp(-\beta) \leq Z \leq m \exp(\beta)$, we have that

$$Z\left(1 - \frac{1}{c}\right) \leq Z - \frac{1}{c}Z^- + \frac{1}{\alpha}\exp(\beta s_i) \leq Z\left(1 + \frac{\exp(2\beta)}{c}\right),$$

and thus by Lemma 7 we have that

$$\frac{1}{Z}\left(1 - \frac{\exp(2\beta)}{c}\right) \leq E\left[\frac{1}{\exp(\beta s_1) + \frac{1}{\alpha}S_{c-1} + \frac{1}{\alpha}\exp(\beta s_i)}\right]$$

$$\leq \frac{1}{Z(1 - \frac{1}{c})} + \frac{1}{\alpha^2}\frac{\mathrm{Var}(S_{c-1})}{m^3\exp(-3\beta)}$$

$$\leq \frac{1}{Z(1 - \frac{1}{c})} + \frac{\exp(5\beta)}{\alpha m^2}$$

$$\leq \frac{1}{Z}\left(1 + O\left(\frac{1}{c}\right)\right) + \frac{\exp(5\beta)}{\alpha m^2}$$

$$= \frac{1}{Z} + \frac{\exp(O(\beta))}{\alpha m^2}.$$

We conclude that

$$E\left[\frac{1}{\exp(\beta s_1) + \frac{1}{\alpha}S_{c-1} + \frac{1}{\alpha}\exp(\beta s_i)}\right] = \frac{1}{Z} \pm \frac{\exp(O(\beta))}{\alpha m^2} \tag{7}$$

Continuing Equation (6) by applying Inequality (7), we have that

$$E\left[\frac{\frac{1}{\alpha}\sum_{j=1}^{c}\exp(\beta s_{k_j})\beta\nabla s_{k_j}}{\exp(\beta s^+) + \frac{1}{\alpha}\sum_{j=1}^{c}\exp(\beta s_{k_j})}\right] = \sum_{i=2}^{m}\left(\frac{\exp(\beta s_i)\beta\nabla s_i}{Z} \pm \frac{\exp(\beta s_{k_j})\beta\nabla s_{k_j}\exp(O(\beta))}{\alpha m^2}\right)$$

$$= \left(\sum_{i=2}^{m}\frac{\exp(\beta s_i)\beta\nabla s_i}{Z}\right) \pm \frac{\exp(O(\beta))}{\alpha m^2}\sum_{i=2}^{m}\exp(\beta s_i)\nabla s_i$$

$$= \left(\sum_{i=2}^{m}\frac{\exp(\beta s_i)\beta\nabla s_i}{Z}\right) \pm \frac{\exp(O(\beta))}{\alpha m^2}\sum_{i=2}^{m}\nabla s_i. \tag{8}$$

Combining Inequalities (5) and (8), we have that

$$E\left[\frac{U}{V}\right] - \frac{E[U]}{E[V]} = \pm\frac{\exp(O(\beta))}{\alpha m^2}\sum_{i=1}^{m}\nabla s_i,$$

and thus

$$\left\|E\left[\frac{U}{V}\right] - \frac{E[U]}{E[V]}\right\|_2 = \frac{\exp(O(\beta))}{\alpha m^2}\sum_{i=1}^{m}\|\nabla s_i\|$$

$$= \frac{\exp(O(\beta))M}{\alpha m}.$$

$\square$

We can now prove Theorem 5.

*Proof.* We use Theorem 2 to bound the bias due to the staleness of the cache and Lemma 4 to bound the bias due to using a sampled cache. We can then apply Lemma 6 to finish the proof. $\square$