# OpenReview forum: "Efficient Training of Retrieval Models using Negative Cache"
_NeurIPS.cc/2021/Conference — NeurIPS 2021 Poster_

### Official Review · Reviewer_bmuN · 2021-07-06

**Rating:** 6
**Confidence:** 3

**Summary:**

This paper is about finding good negative samples for contrastive training. The authors take a caching approach, and show how to make it work in a streaming fashion. Lots of theoretical results are part of the contribution.

**Limitations And Societal Impact:**

No, but I don't believe they need to for this basic topic.

**Main Review:**

EDIT: The authors have added further experiments, so I am raising my score to a 6.

----------

Thanks to the authors for the hard work on this paper, which is well-written and clear. The idea is a nice one, and the descriptions + theory up to the experiments section are excellent.

I am, however, conflicted because the experimental section is very brief and not very insightful. At the end, the paper reads like the authors ran out of space. In such a situation, I would have expected additional experimental results + analyses in the appendix, but that is not the case. There are a number of questions that would be good to answer such as:

- Why does cache (2M) do worse than cache (512k) on TREC?
- What is the difference in training time and overall GPU/CPU memory between various cache sizes vs ANCE? In short, you should add ANCE and DPR to Table 1.
- The ANCE paper has experiments for the retrieval components of Natural Questions and Trivial QA. How does your algorithm compare to ANCE on those datasets?

I think it's important for the paper to have more thorough experiments and analysis to fully convince the reader of its efficacy, and I am very open to increasing my score after some additional experiments are added (in the appendix is OK).

Minor
- Line 181 - "rho" should be $\rho$
- Line 196 - "both" should be "all three"
- Line 294 - "500 thousand" -> "500,000"
- Line 296 - "10(MRR@10)" -> missing space
- Generally writing "query, passage" as "(query, passage)" would make it easier to read.
- Line 311 - "we are interesting in seeing" - "we are interested in seeing"

**Time Spent Reviewing:**

3

---

> ### Author Response · Authors · 2021-08-10
> **Response to Reviewer bmuN**
>
> We thank you for your review.
>
> > Why does cache (2M) do worse than cache (512k) on TREC?
>
> We note that the 2M cache does better in terms of MRR on MS MARCO. MRR is closer to the training objective, as evaluation data is from the same distribution as the training data (click data), while the TREC eval data is a slightly different task (human labeled data).
>
> > What is the difference in training time and overall GPU/CPU memory between various cache sizes vs ANCE? In short, you should add ANCE and DPR to Table 1.
>
> ANCE uses all 8.8M negatives, so it should be comparable to the 90G memory required in the final line of the table. DPR does not use a cache, so it should be a bit under the 15G in the first part of the table. We didn’t include wall time estimates due to the difference in computing resources, however all our experiments complete in less than a day. Reviewer bG1r cites a paper claiming that ANCE can take nearly a month to converge.
>
> > The ANCE paper has experiments for the retrieval components of Natural Questions and Trivial QA. How does your algorithm compare to ANCE on those datasets?
>
> We provide results on two standard benchmarks TREC and MSMARCO. We particularly focused on MS MARCO because it had the largest number of queries  (384,597 compared to 58,880 in the version of the NQ dataset used by DPR), which is good for training transformer models. In the future we will add results on more datasets.

---

> > ### Comment · Reviewer_bmuN · 2021-08-22
> > **Thanks**
> >
> > Thank you for your response.
> >
> > I do understand "the TREC eval data is a slightly different task (human labeled data)" - why does that indicate that the cache 512k is better than the cache 2M?
> >
> > Please consider adding the numbers you quoted about training time and memory to Table 1, or at least a sentence that indicates them.
> >
> > A main reason that my score was a 5 is that results are only for two datasets. Given that you don't plan to add more benchmarks for the current submission, I will keep my score the same.

---

> > > ### Author Response · Authors · 2021-09-01
> > > **Further clarification on Cache 512K vs Cache 2M**
> > >
> > > (1) Following prior work, the TREC eval and the MS MARCO eval are done with the same model parameters trained with the MS MARCO training data. It may be that the larger cache improved the MS MARCO eval at a small expense in the TREC eval, since it improves training with the MS MARCO training data.
> > >
> > > (2) We will update training time and memory in Table 1 in the next version of the paper.
> > >
> > > (3) We have completed experiments with the Natural Questions dataset. Please see our earlier comment for details.

---

> > > > ### Comment · Reviewer_bmuN · 2021-09-01
> > > > **Thank you for further experiments**
> > > >
> > > > The additional experiments are quite convincing. Assuming you will add them to the paper, I am raising my score by 1 point. Thank you again for the extra work.

---

> ### Author Response · Authors · 2021-09-01
> **Additional Experimental Results on the Natural Question Dataset**
>
> We thank you again for your constructive review.
>
> We completed our experiments on the Natural Question dataset (NQ). When we restrict the streaming negative cache algorithm to only 10% of the passage size we achieve a Recall@20 / Recall@100 of 78.4 / 85.6. This compares to 78.4 / 85.5 for DPR, however we use no external negatives such as the BM25 negatives used by DPR and ANCE. The use of BM25 negatives is specific to NLP tasks and requires an additional step of inference over all queries and documents, which we are able to avoid. This further establishes that our method is scalable, generic, and can be extended to very large datasets with minimal resources. We will include this experiment in the next version of the paper.

---

### Official Review · Reviewer_bG1r · 2021-07-15

**Rating:** 7
**Confidence:** 3

**Summary:**

This paper addresses the technique of negative sampling in IR tasks. The author proposes to utilize Gumbel-Max sampling on the cached embeddings to efficiently sample the negatives. For a very large dataset, the paper develops a streaming variant of the algorithm. The author also conducts some theoretical analysis to demonstrate its convergence and bias. Moreover, the experimental results show its efficiency.

**Limitations And Societal Impact:**

No, the conclusion section is also missing.

**Main Review:**

- Originality - The idea of using cached embeddings and nearest neighbors for improving negative mining is recent but not completely new. Previous works are REALM [1] and ANCE [2]. Both articles are acknowledged by authors in this paper. The main contribution of this work concerning previous works is the use of Gumbel-Max sampling to obtain an unbiased estimate of the gradient.

- Quality - The paper aims to develop an efficient way to train dual encoders using a large cache of negative samples, and the paper contains a lot of theoretical demonstrations to analyze the convergence and bias induced by the proposed method.

- Clarity - The paper is well written and the ideas expressed in the paper are well presented. However, it’s better to include recent relevant papers, such as LTRe [3] and DRhard [4] with a static document index. Furthermore, it seems that the conclusion part is missing.

- Significance - The line of work presented in this paper is relevant for dense retrieval. However, experimental results and the analysis of these results seem to be slightly insufficient. I believe authors should compare with some recent works with static document index.

Pros
1. The motivation is clear. This paper proposes a useful method for sampling a huge number of hard negative samples to improve performance.
2. This paper is well-written and easy to follow.
3. The idea of using Gumbel-Max sampling to accelerate training is interesting and experimental results prove its efficiency.
4. The theoretical analysis about convergence and bias is sufficient and solid.

Cons
1. The conclusion section is missing.
2. Results seem discouraging. The proposed cache method (with 2M negatives) cannot outperform ANCE, so I am concerned about the effectiveness.
3. It would be better to cite more recent works with static document indexes and compare them with the cached negative technique.

Important questions/requests:
1. In the abstract, the author claims that their proposed negative cache sampling enables training with reduced memory and computation, but according to table 1 on page 9, when using a cache with 2M elements, it consumes 38G per step, which is huge. So, is it contradictory to what the author said?
2. Please supplement your conclusion.

Typos/Minor questions:
1. Line 14, compares -> compare
2. Both in Equations 1 and 2, the $s_{i y_{i}}$ in the denominator should be $s_{ij}$.
3. Line 81, $s_{ij}$ seems a typo.

[1] Approximate nearest neighbor negative contrastive learning for dense text retrieval, Lee Xiong et al, ICLR-2021.

[2] Realm: Retrieval-augmented language model pre-training, Kelvin Guu et al, Arxiv-preprint.

[3] Learning to Retrieve: How to Train a Dense Retrieval Model Effectively and Efficiently, Zhan, Jingtao, et al. arXiv preprint

[4] Optimizing Dense Retrieval Model Training with Hard Negatives, Zhan, Jingtao, et al. SIGIR 2021


**Time Spent Reviewing:**

8

---

> ### Author Response · Authors · 2021-08-10
> **Response to Reviewer bG1r**
>
> We thank you for your review.
>
> > Results seem discouraging. The proposed cache method (with 2M negatives) cannot outperform ANCE, so I am concerned about the effectiveness.
>
> Our method gives similar performance as ANCE even though it uses a much bigger index than our cache (8.8M vs 2M). We note that our method has stronger theoretical guarantees and saves additional cost of running separate ANN indexing / retrieval processes. Additionally we note that in citation [3] of your review that it mentions that ANCE takes nearly a month to train. Our approach takes less than a day and requires no pre-training beyond the BERT-base model.
>
> > It would be better to cite more recent works with static document indexes and compare them with the cached negative technique.
>
> Thank you for these references. We believe that our work complements LTRe and DRHard, as they fine tune a pre-trained dense retrieval model (ANCE in the papers cited). Our work can be seen as a more scalable way to obtain the initial document encoder. We will include this discussion in the paper.
>
> > In the abstract, the author claims that their proposed negative cache sampling enables training with reduced memory and computation, but according to table 1 on page 9, when using a cache with 2M elements, it consumes 38G per step, which is huge. So, is it contradictory to what the author said?
>
> Our algorithm can scale with the amount of accelerator memory, and use a smaller cache size if less memory is available. Additionally, 38G of memory is not unreasonable, as it can fit in 2 consumer level RTX 3090 GPUs.
>
> > Please supplement your conclusion.
>
> Due to space we were not able to fit a conclusion in the submitted version of the paper. We will add a conclusion to the next version.

---

> > ### Comment · Reviewer_bG1r · 2021-08-25
> > **Thanks**
> >
> > Thank you for your response.
> > Almost all my questions have been addressed.

---

### Official Review · Reviewer_z8bE · 2021-07-17

**Rating:** 7
**Confidence:** 4

**Summary:**

The paper proposes a combination of a Gumbel-Max sampling and embedding caching to reduce the complexity of the negative sampling. The paper has theory and rather convincing results.

**Main Review:**

In the initial review I had a concern that authors need to explain why their work is not directly compared to the RocketQA approach, which achieves a much better accuracy (ok claims to achieve) on the MS MARCO dev data set:

Qu Y, Ding Y, Liu J, Liu K, Ren R, Zhao WX, Dong D, Wu H, Wang H. RocketQA: An optimized training approach to dense passage retrieval for open-domain question answering. In the process of discussion, we agreed that RocketQA approach is relevant, but likely complementary.

However, after the discussion with other reviewers, a bigger concern was that a single data set is not enough to demonstrate the value of their approach. I downgraded the score and recommend some the following:

1. ms marco v2
2. One or more retrieval-QA datasets from facebook DPR paper: https://github.com/facebookresearch/DPR

Since authors ran experiments on the NQ/DPR dataset it's only fair to recommend acceptance of the paper.

**Time Spent Reviewing:**

2

---

> ### Author Response · Authors · 2021-08-10
> **Response to Reviewer z8bE**
>
> We thank you for your review.
>
> > The authors need to explain why their work is not directly compared to the RocketQA approach.
>
> Thank you for the citation, we will include it in the next version of our paper. RocketQA’s main contribution is somewhat orthogonal to ours in that they use a cross-attention model to filter hard negatives and to augment the training data with distillation data. The cross-attention model requires a strong dual encoder model to generate training data. They use 512 cross-replica negatives to train this model. As we see in Table 1 of our paper, even 256 cross-replica negatives requires more memory and has a lower steps / second than our approach. In Table 6 of RocketQA, we see that our model outperforms their corresponding BERT-base model. Our method can be seen as a scalable approach to train the initial dual-encoder model.

---

> ### Author Response · Authors · 2021-09-01
> **Additional Experimental Results on the Natural Question Dataset**
>
> We thank you again for your constructive review.
>
> We completed our experiments on the Natural Question dataset (NQ). When we restrict the streaming negative cache algorithm to only 10% of the passage size we achieve a Recall@20 / Recall@100 of 78.4 / 85.6. This compares to 78.4 / 85.5 for DPR, however we use no external negatives such as the BM25 negatives used by DPR and ANCE. The use of BM25 negatives is specific to NLP tasks and requires an additional step of inference over all queries and documents, which we are able to avoid. This further establishes that our method is scalable, generic, and can be extended to very large datasets with minimal resources. We will include this experiment in the next version of the paper.

---

### Official Review · Reviewer_ChBd · 2021-07-17

**Rating:** 7
**Confidence:** 4

**Summary:**

This paper focuses on the problem of negative mining for training bi-encoders with cross-entropy loss in a retrieval setting.
Overall, I think this is a good paper. The authors provide a method that compares to ANCE on the MS MARCO dataset, but is simpler and much more efficient. The ideas are well argumented, and experiments seem to be consistent with the theoretical aspects. Additional experiments would have made the paper stronger, but results on MS MARCO might be sufficient for acceptance.


**Limitations And Societal Impact:**

Not discussed

**Main Review:**

This paper focuses on the problem of negative mining for training bi-encoders with cross-entropy loss in a retrieval setting. This problem is not new, but has received some attention recently in the NLP/IR community. Strategies like ANCE show significant improvements compared to random/BM25 sampling, at the cost of maintaining a complex training pipeline with asynchronous index refresh. Other simpler strategies like in-batch negatives exhibit limitations (poor negatives).

The authors propose a new method lying between the two types of approaches, based on a large negative cache that is used to sample “good” negatives. They derive a sampling based on the gumbel-max trick to efficiently optimize the cross-entropy loss without needing to embed every document at each iteration, and propose a “streaming” version that relies on a sample multiset of the full collection. At each iteration, a small fraction of documents in the cache are updated, leading to minimal effects on the training speed. The authors give some convergence bounds for both full and streaming versions, and experiments seem to be consistent with them. The method bears similarities with MoCo (using a cache for negatives) but is conceptually different (dual encoder training).

Pros:

- The method compares favorably to ANCE (which is more or less sota on MS MARCO), but is simpler and more efficient.
- The paper is easy to follow in my opinion, and the key messages are clear.
- It is interesting to see that a rather small cache size (128k, so ~1.4% of the collection) is already sufficient to increase the perf by 1 point in MRR@10 compared to DPR.

Cons/clarification:

- Maybe I missed something, but why is the cache a multiset (i.e. allowing repetitions for documents) ? Wouldn’t it be better to sample at each iteration elements which are **not** already in the cache?
- Line 226/227: I do not understand what are 512 vs 1024?
- In Algorithm 2, I do not get the union in the GumbelMaxSample? As you said in 3.3, you do not want $y_i$ to be sampled right ?
- Your approach is rather generic; I would have liked to see experiments at least on another dataset. Do you think the improvements of your approach also reflect some issues with MS MARCO (especially the fact that it contains a lot of false negatives) ? We expect that the approach indeed samples better negatives, but does it also avoid sampling such false negatives ?

Some typos / minor comments:
- Eq (1): y_i => y_j for the denominator
- line 304: scale => scaling ?
- algorithm 1: missing beta
- l. 196: Both of ... which of the three?


**Time Spent Reviewing:**

2.5

---

> ### Author Response · Authors · 2021-08-10
> **Response to Reviewer ChBd**
>
> We thank you for your review.
>
> > Why is the cache a multiset (i.e. allowing repetitions for documents) ? Wouldn’t it be better to sample at each iteration elements which are not already in the cache?
>
> For the proof of Theorem 4, we need the samples from the cache to be i.i.d. In practice, we do sample an element not in the cache. We will clarify this in the next version of the paper.
>
> > Line 226/227: I do not understand what are 512 vs 1024?
>
> These are the input and output dimensions used for MS MARCO training, following the values used by our experiments and ANCE. We use them to give an example calculation of the memory needed for the cache. We will clarify this line.
>
> > In Algorithm 2, I do not get the union in the GumbelMaxSample? As you said in 3.3, you do not want yi to be sampled right ?
>
> Yes, in our experiments we do not allow yi to be sampled. We will modify Algorithm 2 to reflect Section 3.3 and the experiment code.
>
> > Your approach is rather generic; I would have liked to see experiments at least on another dataset.
>
> We provide results on two standard benchmarks TREC and MSMARCO. We particularly focused on MS MARCO because it had the largest number of queries  (384,597 compared to 58,880 in the version of the NQ dataset used by DPR), which is good for training transformer models. In the future we will add results on more dataset.

---

> > ### Comment · Reviewer_ChBd · 2021-09-10
> > **Thanks**
> >
> > For the contribution and your answer, this is a valuable paper for the IR/NLP community I think

---

> ### Author Response · Authors · 2021-09-01
> **Additional Experimental Results on the Natural Question Dataset**
>
> We thank you again for your constructive review.
>
> We completed our experiments on the Natural Question dataset (NQ). When we restrict the streaming negative cache algorithm to only 10% of the passage size we achieve a Recall@20 / Recall@100 of 78.4 / 85.6. This compares to 78.4 / 85.5 for DPR, however we use no external negatives such as the BM25 negatives used by DPR and ANCE. The use of BM25 negatives is specific to NLP tasks and requires an additional step of inference over all queries and documents, which we are able to avoid. This further establishes that our method is scalable, generic, and can be extended to very large datasets with minimal resources. We will include this experiment in the next version of the paper.

---

> > ### Comment · Reviewer_ChBd · 2021-09-10
> > **Great**
> >
> > It would be good to have these new results included in the paper

---

### Author Response · Authors · 2021-09-01
**Additional Experimental Results on the Natural Question Dataset**

We thank all the reviewers again for their constructive reviews.

We completed our experiments on the Natural Question dataset (NQ). When we restrict the streaming negative cache algorithm to only 10% of the passage size we achieve a Recall@20 / Recall@100 of 78.4 / 85.6. This compares to 78.4 / 85.5 for DPR, however we use no external negatives such as the BM25 negatives used by DPR and ANCE. The use of BM25 negatives is specific to NLP tasks and requires an additional step of inference over all queries and documents, which we are able to avoid. This further establishes that our method is scalable, generic, and can be extended to very large datasets with minimal resources. We will include this experiment in the next version of the paper.

---

### Decision · Program_Chairs · 2021-09-27

**Decision:**

Accept (Poster)

**Comment:**

This paper presents a novel and more efficient strategy for approximate negative sampling in IR system training. Theoretical analysis of convergence is included, and empirical results show efficiency gains over the state-of-the-art. All reviewers recommend acceptance, with the majority voting strongly for acceptance. Generally, reviewers praise clarity and motivation, as well as theoretical and empirical results. One reviewer initially had concerns about generalization to additional datasets. This concern was adequately addressed by author response. Overall I agree with reviewers and recommend acceptance.